# MaMMUT: A Simple Architecture for Joint Learning for MultiModal Tasks

**Weicheng Kuo**[*]    **AJ Piergiovanni**[*]    **Dahun Kim**[†]    **Xiyang Luo**[†]    **Ben Cain**    **Wei Li**    **Abhijit Ogale**
**Luowei Zhou**    **Andrew Dai**    **Zhifeng Chen**    **Claire Cui**    **Anelia Angelova**

**Google Research**[§]

**Reviewed on OpenReview:** <https://openreview.net/forum?id=FqOG4osY7C>

## Abstract

The development of language models have moved from encoder-decoder to decoder-only designs. In addition, we observe that the two most popular multimodal tasks, the generative and contrastive tasks, are nontrivial to accommodate in one architecture, and further need adaptations for downstream tasks. We propose a novel paradigm of training with a decoder-only model for multimodal tasks, which is surprisingly effective in jointly learning of these disparate vision-language tasks. This is done with a simple model, called MaMMUT. It consists of a single vision encoder and a text decoder, and is able to accommodate contrastive and generative learning by a novel two-pass approach on the text decoder. We demonstrate that joint learning of these diverse objectives is simple, effective, and maximizes the weight-sharing of the model across these tasks. Furthermore, the same architecture enables straightforward extensions to open-vocabulary object detection and video-language tasks. The model tackles a diverse range of tasks, while being modest in capacity. Our model achieves the state of the art on image-text and text-image retrieval, video question answering and open-vocabulary detection tasks, outperforming much larger and more extensively trained foundational models. It shows very competitive results on VQA and Video Captioning, especially considering its capacity. Ablations confirm the flexibility and advantages of our approach.

## 1 Introduction

Vision-language learning has become critical in improving both visual-understanding and multimodal vision-language tasks. Large foundational vision-language models, which are designed to be extended to multiple downstream tasks, follow two main training strategies, typically exemplified by disjoint architectures. Some vision-language pre-training approaches apply a contrastive loss, in a dual-encoder style architecture, e.g. CLIP, Align, Florence (Radford et al., 2021; Jia et al., 2021; Yuan et al., 2021). Contrastive training has been shown to produce strong backbones, which lead to successful image understanding and cross-modal retrieval tasks, e.g. image-to-text or text-to-image retrieval.

Alternatively, the autoregressive and masked token modeling objectives, well known from language modeling, are very popular with vision-language models for text generation. They are often referred to as split-captioning objectives. The split-captioning training is typically beneficial to text-generative tasks e.g. VQA (Agrawal et al., 2015).

The most common architectures used in these scenarios are the encoder-decoder ones, which use a separate vision and text encoders, or a joint vision-text encoder, before a joint decoder, applying decoding losses from language learning (Cho et al., 2021; Chen et al., 2022; Piergiovanni et al., 2022a; Wang et al., 2021; 2022c). Architectures with cross-attention over frozen or partly frozen language models have also been popular (Alayrac et al., 2022).

Combining these two types of architectures and loss functions has proven to be challenging, with recent approaches such as Align-Before-Fuse (ALBEF) and CoCa (Li et al., 2021; Yu et al., 2022; Yan et al., 2022) requiring multiple components or training stages, and special recipe to accommodate video tasks (Yan et al., 2022). Simultaneously, many pure language-only models have adopted simple decoder-only architectures with great success (Liu et al., 2018; Brown et al., 2020; Du et al., 2021) and with the added benefit of significant parameter reduction (Liu et al., 2018).

---

[§]Correspondence to `{weicheng, ajpiergi, anelia}@google.com`.

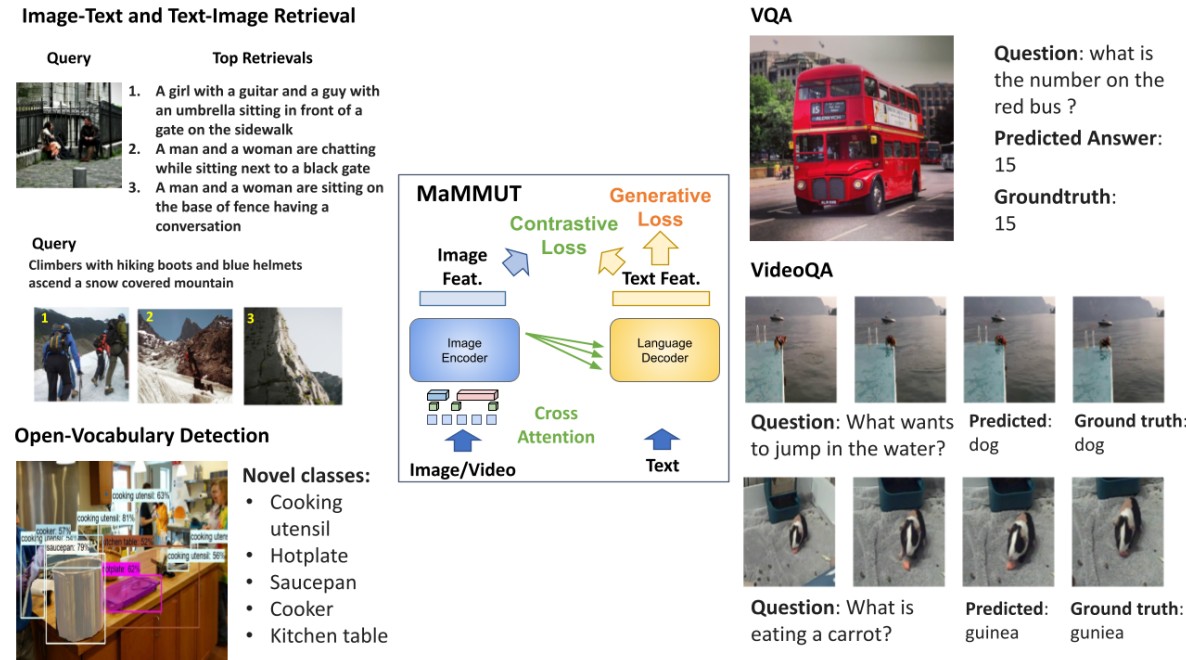

Figure 1: The MaMMUT model is a simple vision-encoder and text-decoder architecture, which serves as foundational model for both image-language and video-language tasks. Despite its relatively small size, the model outperforms SOTA on many diverse tasks. Example results on Image-Text/Text-Image retrieval, Visual Question Answering (VQA), Open-vocabulary detection, Video Question Answering (VideoQA) are shown.

We here propose a simple approach to unify contrastive learning, localization aware, and autoregressive captioning pretraining, by using a single language decoder and an image encoder. Our formulation is more general and allows maximal weight-sharing and parameter efficiency between the contrastive and generative tasks. To address the challenge of reconciling the unconditional sequence-level representation learning needed for contrastive learning with the token-conditioned next-token prediction, we propose a two-pass learning strategy using the text decoder. In one pass of the training, we utilize cross attention and causal masking to learn the caption generation task, where the text features can attend to the image features and predict the tokens in sequence; in the other pass we disable the cross-attention and causal masking, which learns the contrastive task without visibility into the image features. We further modify the contrastive training objective to be localization-aware, further equipping the model for object detection tasks. With this training strategy our model can address a diverse range of tasks, e.g. retrieval, text generative or detection. This provides a simpler alternative to previous approaches (Li et al., 2021; Yu et al., 2022; Singh et al., 2022; Yuan et al., 2021), where our model architecture is more compact and shared more broadly across tasks in the model. Furthermore, our model allows us to apply this architecture to video by a seamless adaptation based on the TubeViT approach (Piergiovanni et al., 2023a). This allows us to address video-text tasks, such as Video Question Answering (VideoQA) and Video Captioning successfully, outperforming prior large image-text or video foundatinoal models, such as 80B Flamingo, by pre-training on image-and-text dataset only. Furthermore, we extend the approach to leverage the pretrained model for open-vocabulary detection, demonstrating the localization capabilities of the model. We call the model MaMMUT.

Our model is simple, but at the same time represents a union of tasks that other models find it challenging to put together, or need more specialized adaptations for. In comparison, many of the prior approaches, despite reporting results on 10, 20 or more tasks or training bigger models (Chen et al., 2022; Singh et al., 2022; Alayrac et al., 2022; Yuan et al., 2021), accommodate fewer categories of tasks than our model does. One of the largest models, PaLI (Chen et al., 2022) addresses only classification, VQA and image captioning task categories, but is not designed to handle retrieval, detection, or video tasks. GIT (Wang et al., 2022b) extends the model to videos, but does not handle retrieval or detection. FLAVA (Singh et al., 2022), Florence (Yuan et al., 2021), ALBEF (Li et al., 2021), BLIP (Li et al., 2022a) which are based on constrastive learning, accommodate more tasks, but do not perform video-language or object localization tasks, or need significant downstream adaptations to do so (Yuan et al., 2021). Flamingo (Alayrac et al., 2022) is not able to do retrieval or object detection. With MaMMUT, we address all these tasks.

Experiments on image-text, text-image retrieval, VideoQA and Open-Vocabulary Object Detection show above-SOTA performances, outperforming much larger models. MaMMUT shows very competitive performance in Video Captioning and Visual Question Answering (VQA), which is done in the challenging open-ended generation setting. We note that our pre-training is done on image-text noisy pairs only and does not include video or labeled object localization information or supervised labels. Ablation studies confirm the challenges and benefits of developing a simple and flexible model to unify multiple tasks.

Our contributions are: (i) a simple, compact and extensible foundational vision-language model which addresses multiple diverse multimodal tasks, such as image/text retrieval, Visual Question Answering, Open-Vocabulary Detection, Video Question Answering and Video Captioning; (ii) a novel two-pass learning method which trains jointly reconciling the unconditional sequence-level representation learning needed for contrastive learning with the token-conditioned captioning-like learning, using a single shared decoder-only model; (iii) a seamless extension to video tasks with no additional pre-training needed, creating a powerful video-language model.

## 2   Related Work

Vision and language pretraining has gained considerable popularity, where many successful and wide-ranging approaches have been created. Following token masking or next-token prediction losses used in text modeling, image-language pre-training methods extended these ideas for image and text inputs, where a language modeling loss is applied to a model which considers image inputs (Wang et al., 2022b; Chen et al., 2022; Wang et al., 2021; 2022e; Chen et al., 2020b; Zhou et al., 2020; Zhang et al., 2021a; Tan & Bansal, 2019; Li et al., 2020; Lu et al., 2019; Weijie Su, 2020; Dou et al., 2022). This technique enables text-generative vision-and-language tasks, such as VQA, Image Captioning or classification. Extensions to these approaches, where masking is done in the image space (Bao et al., 2021), have also been considered. Some of the above-mentioned approaches also feature similar architecture as ours, with a simple decoder e.g. GIT (Wang et al., 2022b), but the tasks addressed are limited to the text generative ones.

Contrastive vision and language pretraining has been popularized by the CLIP and Align models (Radford et al., 2021; Jia et al., 2021), demonstrating that contrastive learning can produce powerful embedding for many downstream applications. While the contrastive loss has been widely applicable, e.g. in self-supervised learning (Chen et al., 2020a), the appeal of these methods is the ability to learn across two modalities and produce high quality embeddings from large amounts of noisy image-language pairs which are harvested automatically from the web. Many generative pre-training approaches have also subsequently leveraged such noisy and easily obtainable datasets for vision-language pre-training purposes. Contrastive vision-language models are typically two-tower models, where nontrivial modifications are needed for downstream tasks (Yuan et al., 2021).

Several prior works proposed approaches to combine contrastive and generative vision-language pre-training, on the premise of two-tower models and cross-attention or cross-modal masking to align the modalities (Li et al., 2021; Singh et al., 2022; Li et al., 2022a; 2023; Yu et al., 2022). ALBEF (Li et al., 2021) applies a contrastive loss to an image and text encoder models and adds a decoder for generative tasks. BLIP-2 (Li et al., 2023) leverages an off-the-shelf frozen image encoder and large language model for generative learning. Similar to ALBEF (Li et al., 2021), CoCa (Yu et al., 2022) uses a language generation rather than masked language modeling objective. Furthermore, a second decoder uses cross-attention to connect image representations with text to generate the output text. In these approaches there is a distinction of unimodal vs multimodal text model, and contrastive loss is only applied to the unimodal part. Compared to CoCa, our approach differs in a few aspects: ours is a fully shared text model enabled by the proposed two-pass learning paradigm; it achieves state-of-the-art performance on video and open-vocabulary detection tasks which are not easily derived from language generation; has affordable computational cost for reproducibility; in terms of zero-shot performance, it has state-of-the-art zero-shot image-text retrieval. Other approaches (Wang et al., 2022d; Lu et al., 2022) unify understanding and generation tasks in the same sequence-to-sequence framework, spanning a larger number of tasks, including tasks, such as, referring expressions (Yu et al., 2016). BEiT-3 (Bao et al., 2021) learns unimodal visual representation from image token reconstruction and finetunes the backbone for downstream tasks.

A number of video foundational models have been proposed (Zellers et al., 2021; Fu et al., 2021; Alayrac et al., 2022; Wang et al., 2022f;c; Cheng et al., 2022; Luo et al., 2020). The Flamingo model (Alayrac et al., 2022) extends a large frozen language model, with image and video inputs to deliver impressive results. VIOLET (Fu et al., 2021) uses masked language and masked-video modeling for joint video-text learning. Other approaches extend a pre-trained

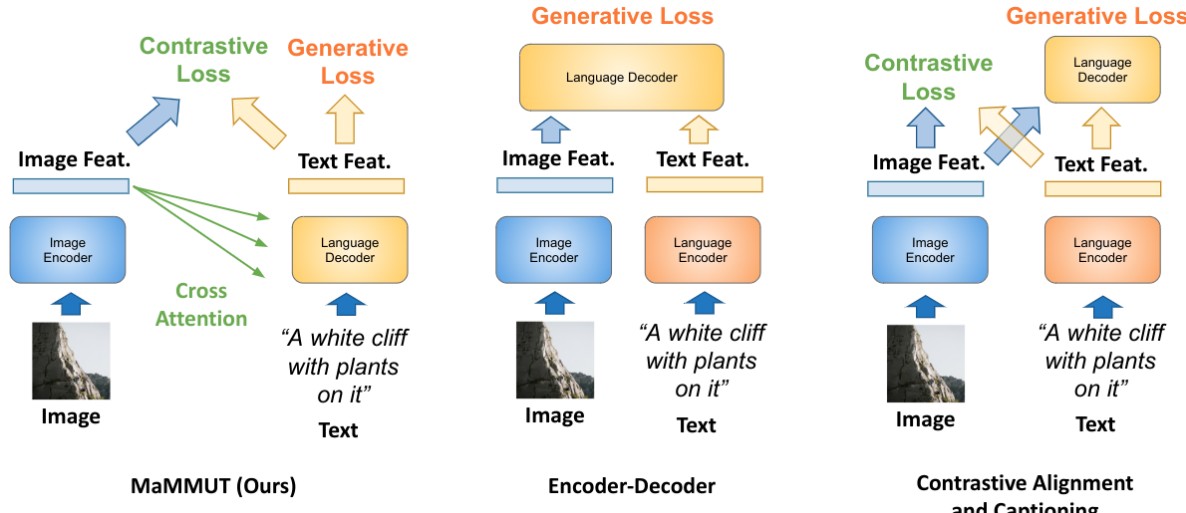

Figure 2: MaMMUT model architecture with an image encoder and text decoder (left), compared to others. Many encoder-decoder architectures (center) cannot handle the contrastive objective, for example (Cho et al., 2021; Chen et al., 2022). Approaches to combine contrastive and captioning (right), e.g. Align-Before-Fuse (ALBEF) (Li et al., 2021) or CoCa (Yu et al., 2022) develop more complex models and are hard to extend to video inputs or localization tasks. Our architecture is simpler than previous approaches and is able to accommodate more tasks.

image-language model, adapting it to video (Wang et al., 2022b; Yan et al., 2022; Piergiovanni et al., 2023b), where a common approach is to just accumulate features from individual frames of the video. Our model is mostly aligned to the latter class of models, however, we instead directly process the spatio-temporal information of the video and do not need to process the model by a single frame at a time. Our approach is more advantageous, as much of the temporal information is lost in the sinlge-frame approaches. Florence (Yuan et al., 2021) does an adaptation with 3D kernels to the SWIN transformer (Liu et al., 2021), and OmniVL (Wang et al., 2022c) uses the Timesformer model (Gedas Bertasius, 2021), which preserve the temporal information, however the adaptation is generally more complex. Combining contrastive and captioning losses in video is also a popular technique. For example, InternVideo (Wang et al., 2022f) proposed a combination of a masked video encoder-decoder and a cross-modal constrastive dual encoder in a video foundational model. In contrast, while we perform only a light fine-tuning over a image-language pre-trained model, our approach outperforms the above-mentioned more sophisticated and better-trained video models.

Object detection is available in some foundation models, for example VinVL (Zhang et al., 2021a) pre-train the vision-language model in order to detect objects and attributes. Florence (Yuan et al., 2021), adapt vision-language pre-trained models to object detection tasks. At the same time, Open-Vocabulary Detection, which aims at detecting novel categories of objects, is often overlooked by pre-trained vision-language models, where previous approaches focused on detecting objects in a set of pre-defined classes. VILD (Gu et al., 2022), demonstrate that large vision-language models are advantageous for detecting novel object categories. Inspired by this work, while taking a different approach, we show that open-vocabulary detection is an easy extension to our model with strong performance.

## 3 Method

We introduce our model, called MaMMUT, which offers a simple architecture consisting of a single image-encoder and a text-decoder (Figure 2, left). Our model combines the strengths of contrastive learning and autoregressive next-token prediction in a more flexible architecture than existing works. For example, the model is able to train and perform inference for those tasks with the same shared set of weights. Furthermore, one of the advantages of this simple model is that it allows simple extensions to video where the exact same architecture is able to consume directly video features. This is in contrast to prior work which adapt to video by processing individual frames or by more complex mechanisms. Other important downstream tasks, such as open-vocabulary detection, can also be easily added to this model.

### 3.1 MaMMUT Architecture

MaMMUT is an intuitive and simple architecture for multimodal tasks, which consists of a single vision-encoder and a single text-decoder (see left of Figure 2). We encode the images into latent representation using a neural network

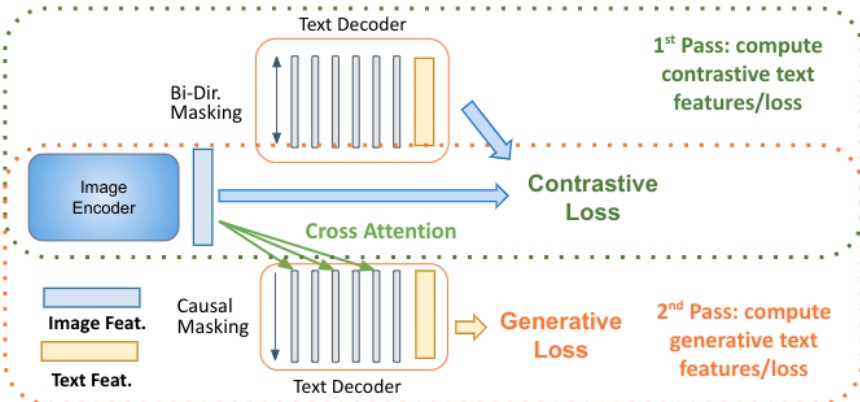

Figure 3: MaMMUT two-pass learning. We compute the image features via the image encoder. Then we compute the contrastive text features and loss by applying a non-causal masking to the language decoder, which makes it effectively a text encoder (top). Finally we compute the generative text features and loss by applying causal masking and cross attention with the image features (bottom). The decoder is visualized twice for clarity, its weights are fully shared.

encoder (such as ViT (Dosovitskiy et al., 2020)). The texts are both encoded and decoded with a Transformer-based decoder-only model. The MaMMUT architecture utilizes cross attention to fuse visual representation with text features anywhere in the decoder layers. This allows the whole decoder to produce both unimodal text representations (needed for a forward pass for contrastive learning) and multimodal text representation (needed to fuse the visual features with the text ones for vision+text tasks). This is done by a two pass joint training and we notably use a single combined training objective (shown later in Equation 5). Below we delve deeper into different aspects of the MaMMUT model.

**Decoder-only Two-Pass learning.**    The main challenge to unify contrastive learning and next-token prediction is to unify the text representation, because the contrastive learning uses unconditional sequence-level representation, whereas captioning optimizes the likelihood of each token conditioned on the previous tokens.

We propose a two-pass approach to jointly learn the two types of text representations by the same model. During the first pass, to learn the contrastive task, we enable bi-directional masking within the decoder. The text features should not see the image features (which characterizes dual-encoder contrastive learner), but can attend to all tokens at once to produce the sequence-level representation. On the second pass, we utilize cross attention and causal masking to learn the caption generation task. The text features can attend to the image features and predict the tokens in a sequence (see Figure 3). As mentioned on the first pass we disable the cross-attention and causal masking in order to separate the text and image features for contrastive learning. All text-decoder weights are shared and gradients are aggregated from the two passes during training. The two passes are done interchangeably during training so their order is not important. As the two passes can share identical image representation, MaMMUT achieves considerable computation savings compared to training two separate models.

We use a vision transformer as image encoder, and project the output dimension of image encoder to that of text decoder by a linear layer. The global image ($v_i$) and text representation ($l_i$) for contrastive learning are computed by average pooling over spatial dimensions and the sequence length, respectively. We insert $M$ cross-attention layers into $N$ text decoder layers, where $M \approx \frac{N}{2}$. The ratio $\frac{M}{N}$ represents a trade-off between model capacity and text-generative capability, where higher $M$ tends to benefit the text-generation tasks.

Somewhat surprisingly, MaMMUT requires no task-specific predictor heads to unify the two seemingly disparate tasks other than a vocabulary embedding layer to map the decoder output features to text tokens. The same decoder features are average-pooled to represent the whole sequence ($l_i$). This is verified in our ablations as well. Compared to contrastive captioner (Yu et al., 2022), MaMMUT allows more flexible fusion of image and text features anywhere in the text decoder, and fully share the text-decoder parameters between contrastive and caption generation tasks.

**Model simplicity.** In contrast to other architectures, Figure 2, middle and right, we use a single vision encoder and a single text decoder. Unlike encoder-decoder models (Chen et al., 2022; Wang et al., 2022b;e) and contrastive alignment and captioning (Li et al., 2021; Yu et al., 2022), this is simpler and more flexible architecture. While using an encoder-decoder is popular, it comes with some disadvantages. For example, a pretrained visual encoder is required, and contrastive tasks are usually not feasible. Contrastive alignment and captioning methods extend the encoder-decoder models to handle retrieval, but that mechanism is not easy to adapt to video and creates a hard loss-balancing task.

MaMMUT on the other hand works well for video-text tasks (Sec. 4) and can be seamlessly adapted without extending the architecture, adding more encoders or having to use an encoder multiple times.

## 3.2 Pretraining Losses

Our model combines the strengths of popular vision-language pre-training losses, as are specifically described below, including with our modifications.

**Contrastive Loss.** Compared to the fully supervised classification pretraining, the contrastive learning approach uses separate image and text encoders to compute image and text embeddings of typically web-scale image-text pairs (Radford et al., 2021; Jia et al., 2021; Yu et al., 2022; Zhai et al., 2022). This allows to model to learn from the richer supervisory signal of free-form text than a fixed label set. The two encoders are trained jointly to minimize the contrastive objective (Oord et al., 2018; Radford et al., 2021; Jia et al., 2021). As a result of learning from free-form text, the representation is very effective for zero-shot image classification, image-text retrieval, and robust to corrupted or out-of-distribution images (Radford et al., 2021).

**Image Captioning loss.** Dual-encoder contrastive models treat the text as a single entity to be encoded, while the encoder-decoder image captioner seeks a more detailed token-level understanding by predicting each word in sequence (see middle of Figure 2). This is achieved through an encoder-decoder architecture, where the image encoder creates a latent representation of the image using ViT (Dosovitskiy et al., 2020) or ConvNet backbone. The text decoder then generates the tokens autoregressively by maximizing the likelihood of each predicted token given the previously generated tokens in the sequence, resulting in the following forward autoregressive formulation:

$$L_{captioning} = -\sum_{t=1}^{T} \log P_\theta(y_t | y_{1,2,...,t-1}, x). \tag{1}$$

To achieve maximum learning efficiency and parallelize computation, the encoder-decoder architecture is trained using a technique called teacher-forcing, which trains the model to predict the tokens at all time steps in parallel.

**Focal Contrastive Loss.** Contrastive learning typically relies on large batch size to extract supervisory signal from noisy image-text data. Our goal is to learn from the more informative and challenging examples than what is possible with the standard cross entropy loss. The focal loss (Lin et al., 2017) presents a compelling alternative as it allows us to finely tune the weights assigned to challenging examples, demonstrating improved performances for object classification or detection scenarios. It has been recently shown, that applying focal loss achieves very competitive performance with significantly smaller batch size for contrastive learning (Kim et al., 2023). More specifically the focal loss is applied as follows: Let $v_i$ and $l_i$ be the normalized image and text embeddings, and the image-to-text (I2T) focal contrastive loss be $L_{\text{focal}}$. We can write $L_{\text{focal}}$ mathematically as:

$$L_{\text{focal\_contrastive}} = -\frac{1}{B} \sum_{i=1}^{B} \sum_{j=1}^{B} (1 - p_i)^\gamma \log(p_i), \tag{2}$$

where $p_i$ denotes the true class probability as below:

$$p_i = \begin{cases} \sigma(v_i l_j / \tau) & \text{if } i = j \\ 1 - \sigma(v_i l_j / \tau) & \text{if } i \neq j \end{cases} \tag{3}$$

Here $\sigma$ denotes the sigmoid function, $v_i$ and $l_i$ the normalized image and text embeddings, and $\tau$ the learnable temperature to scale the logits. The loss is summed over the number of elements in the batch $B$. where . For simplicity, we use the non-alpha-balanced focal loss (Lin et al., 2017). The total loss is the sum of I2T and T2I losses as follows:

$$L_{contrastive} = L_{\text{I2T}} + L_{\text{T2I}}. \tag{4}$$

The focal contrastive loss modification provides additional sensitivity to objects, which equips the model for downstream detection tasks so we prefer it to the classical contrastive loss in our model.

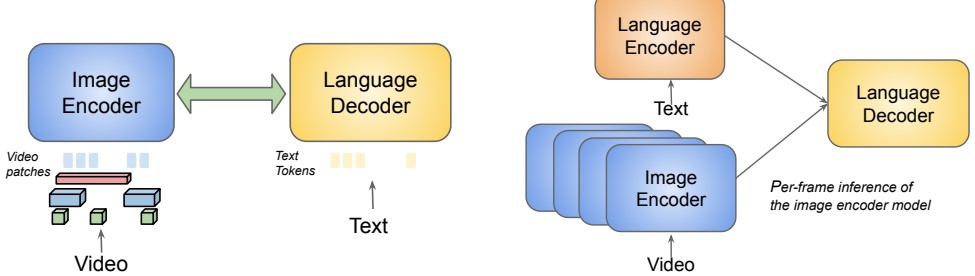

Figure 4: MaMMUT video model (left) efficiently and seamlessly extends the image-language model, by adding learnable spatio-temporal features. The model is applied only once, as opposed to other models, processing each individual frame independently (right). Our model uses image-text pre-training only.

**Final loss**     Combining the losses above, our final loss is as follows. We note that while balancing of the losses is needed as is typical, the losses are simply combined and are shared with the full model.

$$L_{total} = \lambda_{cap}L_{captioning} + \lambda_{focal}L_{\text{focal\_contrastive}}, \tag{5}$$

where $\lambda_{cap}, \lambda_{focal}$, are loss balancing hyper-parameters.

**Learning from Scratch with Noisy Image-Text Supervision.**     Unlike many existing methods that train model components in multiple stages using different data sources or modalities (Chen et al., 2022; Alayrac et al., 2022; Wang et al., 2022e;b), MaMMUT is pretrained end-to-end from scratch, without relying on any prior training or external sources. We use only a web alt-text dataset (Jia et al., 2021) for training. As the image encoder is typically the computation bottleneck in contrastive learning (Radford et al., 2021), our pretraining approach incurs only a relatively light overhead in training efficiency over a pure contrastive learner (Radford et al., 2021) ($\approx 16\%$). This is highly desirable as scaling up model and data size have shown consistent benefits in contrastive and captioning pretraining (Radford et al., 2021; Chen et al., 2022; Alayrac et al., 2022).

**Learned Positional Embeddings for Localization Awareness.**     Existing vision and language pretraining approaches and detection finetuning have a mismatch in how they use positional embeddings. Pretraining approaches typically use full-image positional embeddings during training and apply the same embeddings for downstream tasks. But for detection finetuning, recognition occurs at the region level, requiring the full-image positional embeddings to generalize to regions not seen during pretraining. To address this gap, we adopted the Cropped Positional Embedding (Kim et al., 2023). The idea is to up-sample the positional embeddings from the pretraining image size (e.g., 224) to the detection task image size (e.g., 1024). Then, a randomly cropped and resized region from the up-sampled positional embeddings is used as the image-level positional embedding during pretraining. This method trains the model to view each image not as a full image, but as a region crop from a larger, unknown image, which better matches the downstream use case of detection where recognition occurs at the region level instead of the image level.

**Implementation Details.**     Our model consists of a standard ViT-Huge (Dosovitskiy et al., 2020; Xiaohua Zhai & Beyer, 2022) image encoder (of 650M parameters) and a transformer text decoder of 1B parameters. The cross attention layers are applied every two decoder layers, where the model and hidden dimension follow the text decoder (420M params). Ablation results are conducted with ViT-Base image encoder and a smaller text decoder using 128M parameters, and a smaller 4K batch size. The optimal hyper-parameters are then used to run the large model. The large model is trained for 500K steps using a batch size of 16K. We use AdamW optimizer with weight decay value 0.01. Our initial learning rate is 0.001, and both generative and contrastive loss weights are set to 1.0. We first resize every image to 272x272 and randomly crop a 224x224 patch out for pretraining. We apply 10K warmup steps before applying linear LR decay to the end of training. The temperature in contrastive learning is learnable and initialized to 1.0. We use the standard Sentencepiece tokenizer and set the text length to 64 following existing works (Jia et al., 2021; Yu et al., 2022). We use an alt-text dataset of 1.8B image-text pairs (Jia et al., 2021), as is common in prior methods. The dataset is used for both contrastive and generative pretraining. To better match the downstream detection task, we finetune the model for 100K iterations with the Cropped Positional Embedding for further localization awareness.

| Method | image model size | MS COCO (5K test set) | | | | | | Flickr30K (1K test set) | | | | | |
|---|---|---|---|---|---|---|---|---|---|---|---|---|---|
| | | image-to-text | | | text-to-image | | | image-to-text | | | text-to-image | | |
| | | R@1 | R@5 | R@10 | R@1 | R@5 | R@10 | R@1 | R@5 | R10 | R@1 | R@5 | R@10 |
| CLIP (Radford et al., 2021) | 302M | 58.4 | 81.5 | 88.1 | 37.8 | 62.4 | 72.2 | 88.0 | 98.7 | 99.4 | 68.7 | 90.6 | 95.2 |
| ALIGN (Jia et al., 2021) | 408M | 58.6 | 83.0 | 89.7 | 45.6 | 69.8 | 78.6 | 88.6 | 98.7 | 99.7 | 75.7 | 93.8 | 96.8 |
| FLAVA (Singh et al., 2022) | 86M | 42.7 | 76.8 | - | 38.4 | 67.5 | - | 67.7 | 94.0 | - | 65.2 | 89.4 | - |
| FILIP (Yao et al., 2021) | 302M | 61.3 | 84.3 | 90.4 | 45.9 | 70.6 | 79.3 | 89.8 | 99.2 | 99.8 | 75.0 | 93.4 | 96.3 |
| Florence (Yuan et al., 2021) | 637M | 64.7 | 85.9 | - | 47.2 | 71.4 | - | 90.9 | 99.1 | - | 76.7 | 93.6 | - |
| CoCa-L (Yu et al., 2022) | 303M | 65.4 | 85.6 | 91.4 | 50.1 | 73.8 | 81.8 | 91.4 | 99.2 | **99.9** | 79.0 | 95.1 | 97.4 |
| CoCa (Yu et al., 2022) | 1B | 66.3 | 86.2 | 91.8 | 51.2 | 74.2 | 82.0 | 92.5 | 99.5 | **99.9** | 80.4 | 95.7 | 97.7 |
| **MaMMUT (ours)** | 630M | **70.7** | **89.1** | **93.7** | **54.1** | **76.8** | **84.2** | **94.9** | **99.5** | **99.9** | **82.5** | **96.0** | **98.0** |

Table 1: **Zero-shot image-text retrieval results on COCO and Flickr30K benchmarks (dual-encoder models).** We evaluate our pretrained model compared to other methods. We achieve state-of-the-art results by a margin the on image-to-text / text-to-image retrieval benchmarks with comparable model capacity. (**bold**: best).

## 4 MaMMUT for Video Tasks

Our video model is an efficient and seamless extension to the main image-language model, based on the TubeViT idea (Piergiovanni et al., 2023a). It extracts video tubes which are then projected to patches similar to 2D image projections (Figure 4, left). The model is applied only once. Some other image-language models adapted to video by processing each individual frame by the image encoder, e.g. (Wang et al., 2022b; Piergiovanni et al., 2023b; Yan et al., 2022) (Figure 4, right). This is a limitation as only a relatively small number of frames can be processed due to memory and runtime constraints. There is also some evidence that for a large number of frames the number of effective tokens becomes too large which leads to deteriorated performance (Piergiovanni et al., 2023a).

We follow the TubeViT approach (Piergiovanni et al., 2023a). However, we apply several changes. One main challenge is that TubeViT requires fixed position embedding, whereas this does not match well with the learned positional embeddings of the main encoder. Since the video tubes are sparse and can overlap, TubeViT found that the fixed position embeddings were important. To enable those here, we propose using both position embeddings, and adding a weighted connection to the newly added fixed embeddings. Next, we use the same 2D patches, but at a sparse temporal stride, and finally add the tube projections following the settings used in TubeViT. All these tokens are concatenated together, and passed through the shared ViT backbone. Our TubeViT adaptation to video is very lightweight and no additional components or losses are needed. In comparison, previous approaches tend to exhibit higher complexity. InternVideo (Wang et al., 2022f) for example, supports a video masked encoder for MAE (He et al., 2022) losses in addition to a module similar to ALBEF. Flamingo (Alayrac et al., 2022) is able to process either image and video inputs, but it trains a separate image-encoder, the text part of which is discarded when integrating in the main model. We note that we outperform these in our experiments on video tasks.

We note that this change does not need any additional video data pre-training, and all experiments conducted in the paper are done by directly fine-tuning of the MaMMUT image-text model on a video dataset.

## 5 Experiments

### 5.1 Zero-Shot Image Retrieval Results

As Zero-shot image retrieval is a good indicator of the capability of the model without fine-tuning, we first evaluate the performance of MaMMUT on Zero-shot image-text retrieval tasks. Table 1 shows the image-to-text and text-to-image results, compared to the SOTA methods on two popular retrieval benchmarks MS COCO (Chen et al., 2015) and Flickr (Plummer et al., 2015). As established by previous approaches, we evaluate both image-to-text and text-to-image retrieval following the same evaluation protocol and compare with other published dual-encoder models. Our approach significantly outperforms the state-of-the-art on both image-to-text and text-to-image retrieval by 2-4 points on the recall@1 metrics. We note that these are challenging benchmarks many approaches have been evaluated on.

### 5.2 Visual Question Answering Results

We report the performance on the VQAv2 benchmark (Agrawal et al., 2015) in Table 2. Inspired by recent VQA approaches (Chen et al., 2022; Alayrac et al., 2022; Piergiovanni et al., 2022a), we conduct the experiments in the open-ended text generative setting using an English vocabulary size of 256K. Most prior approaches (Wang et al.,

| Method | Test-Dev | Test-Std |
|---|---|---|
| FLAVA (Singh et al., 2022) | 72.8 | - |
| METER (Dou et al., 2022) | 77.7 | 77.6 |
| Unified-IO (Lu et al., 2022) | 77.9 | - |
| OmniVL (Wang et al., 2022c) | 78.3 | 78.4 |
| Florence (Yuan et al., 2021) | 80.2 | 80.4 |
| SimVLM (Yu et al., 2022) | 80.0 | 80.3 |
| OFA (Wang et al., 2022d) | 82.0 | 82.0 |
| CoCa (Yu et al., 2022) | 82.3 | 82.3 |
| BEiT-3 (Yu et al., 2022) | 84.2 | 84.0 |
| ALBEF (Li et al., 2021) | 75.8 | 76.0 |
| AnswerMe (Piergiovanni et al., 2022a) | 73.6 | - |
| BLIP (Li et al., 2022a) | 78.2 | 78.3 |
| GIT (Wang et al., 2022b) | 78.6 | 78.8 |
| Flamingo-80B (Alayrac et al., 2022) | 82.0 | 82.1 |
| BLIP-2-7B (Li et al., 2023) | 82.3 | - |
| PaLI-3B (Chen et al., 2022) | 79.3 | - |
| PaLI-15B (Chen et al., 2022) | 80.8 | - |
| PaLI-17B (Chen et al., 2022) | 84.3 | 84.3 |
| **MaMMUT (2B)** | 80.7 | 80.8 |

Table 2: **Visual Question-Answering (VQAv2).** We benchmark the performance in an open-ended generation setting. Approaches that perform VQA in closed-vocabulary settings are marked in gray. MaMMUT is very competitive among existing open-ended generation methods given its modest model capacity.

| Overall | Yes/No | Number | Other |
|---|---|---|---|
| 80.84 | 93.41 | 63.89 | 73.78 |

Table 3: **Visual Question-Answering (VQAv2) analysis.** Performance by question types on the test-std split. "Yes/No" questions perform best, whereas questions about how many objects are present, are the most challenging.

2021; Yu et al., 2022; Wang et al., 2022e;d) address the VQA task in the classification setting where the best answer is selected from a predefined set of answers (typically of size 3K). Some recent works (Li et al., 2021; 2022a) train the model in an open-ended settings but restrict the decoder to generate only the 3K candidate answers during inference. In contrast, we allow the decoder to use the whole vocabulary during inference. The VQA-as-open-generation setting poses two key challenges: firstly, the produced text must precisely match the expected answer in order to be considered correct, and secondly, our vocabulary size is much larger than the ones utilized in the classification settings.

MaMMUT achieves 80.8 accuracy on this benchmark, which is very competitive among the open-ended generation approaches. For example, MaMMUT outperforms the PaLI-3B by 1.4 points, while using 1.5x fewer parameters (2B total params). Compared to the PaLI-15B, ours achieves the same performance while being 7.5x smaller. In addition, Flamingo and PaLI use a combination of interleaved image-text, alt-text, human-annotated data sources for training (Alayrac et al., 2022; Chen et al., 2022), whereas we use only an alt-text dataset (Jia et al., 2021). Larger models, such as PaLI-17B, outperform ours. Table 3 shows MaMMUT performance on VQA test-std set by question types. MaMMUT performs the best on yes/no question type, and less well on questions that require counting.

### 5.3 Video Question Answering Results

In this section we present the results of our model on the Video Question Answering task, which is a challenging task answering questions about activities, events, objects, or repetition counting within a video. Our VideoQA results are obtained using the image-text pre-trained model, namely we directly fine-tune on a VideoQA dataset without any video-text pre-training. This is in contrast with other works which use video data pre-training, and indicates that our model is already very strong without additional video-text pre-training.

The results are presented in Table 4a, comparing to the state of the art on MSRVTT-QA (Jun Xu & Rui, 2016) and MSVD-QA (Xu et al., 2017) datasets. MaMMUT outperforms the best SOTA approaches, among which are both video foundational models, e.g VIOLET, MERLOT, InternVideo, image-text models adapted to video e.g. GIT and GIT2 (Wang et al., 2022b), and large vision-language models, such as Flamingo (Alayrac et al., 2022). We note that,

similar to the VQA results, our VideoQA results are conducted in the more challenging open-ended generation setting. Our model is 2.5 times smaller than GIT2 (5B parameters), and 40 times smaller than Flamingo (80B parameters).

## 5.4 Video Captioning Results

Video captioning results on the MSRVTT (Jun Xu & Rui, 2016) and MSVD (Chen & Dolan, 2011; Xu et al., 2017) datasets are presented in Table 4b. The results here too are obtained with image-text pre-training only. Our approach performs well related to SOTA. It outperfoms prior approaches on the MSVD benchmark by large margins, and outperforms most others with the exception of GIT/GIT2 on the MSRVTT video captioning dataset.

| Method | MSRVTT-QA | MSVD-QA |
|---|---|---|
| Just Ask (Yang et al., 2021) | 41.5 | 46.3 |
| MERLOT (Zellers et al., 2021) | 43.1 | - |
| OmniVL (Wang et al., 2022c) | 44.1 | 51.0 |
| VindLU (Cheng et al., 2022) | 44.6 | - |
| Iterative Co-Tok (Piergiovanni et al., 2022b) | 45.7 | 48.8 |
| All-in-one (Wang et al., 2022a) | 46.8 | 48.3 |
| Video-Coca (Yan et al., 2022) | 46.3 | 56.9 |
| VIOLET (Fu et al., 2021) | 43.9 | 47.9 |
| VIOLETv2 (Fu* et al., 2023) | 44.5 | 54.7 |
| Dynamic Pretr. (Piergiovanni et al., 2023b) | 45.1 | 47.1 |
| GIT (Wang et al., 2022b) | 43.2 | 56.8 |
| GIT2 (Wang et al., 2022b) | 45.6 | 58.2 |
| InternVideo (Wang et al., 2022f) | 47.1 | 55.5 |
| Flamingo (Alayrac et al., 2022) | 47.4 | - |
| **MaMMUT (ours)** | **49.5** | **60.2** |

(a) **Video QA Results.** MaMMUT outperforms the SOTA on both MSRVTT-QA and MSVD-QA datasets. We note that image-language pre-training is the only one used here. MaMMUT outperforms video-first models e.g. VIOLET, InternVideo, image+video models, e.g. Flamingo, and large pre-trained image-text models adapted to video, e.g. GIT2.

| Method | MSRVTT | MSVD |
|---|---|---|
| ORG-TRL (Zhang et al., 2020) | 50.9 | 95.2 |
| OpenBook (Zhang et al., 2021b) | 52.9 | - |
| SWINBert (Lin et al., 2022) | 53.8 | 120.6 |
| VIOLETv2 (Fu* et al., 2023) | 58.0 | 130.2 |
| MV-GPT (Seo et al., 2022) | 60.0 | - |
| Vid2Seq (Yang et al., 2023) | 64.6 | 146.2 |
| Video-Coca (Yan et al., 2022) | 73.2 | - |
| GIT (Wang et al., 2022b) | 73.9 | 180.2 |
| GIT2 (Wang et al., 2022b) | **75.9** | 185.2 |
| **MaMMUT (ours)** | 73.6 | **195.6** |

(b) **Video Captioning Results.** MaMMUT performs well on both MSRVTT and MSVD Video Captioning Benchmarks, outperforming SOTA on MSVD by large margins. CIDEr scores are shown. As for VideoQA experiments, we use only image-language pre-training, and directly fine-tune the model on each dataset.

Table 4: Video Question Answering (VideoQA) and Video Captioning results.

## 5.5 Open-vocabulary Detection Results

The Open-Vocabulary detection task refers to the ability to detect and name objects (providing bounding boxes) of categories that are unknown to the model. We evaluate our work on the challenging LVIS dataset (Gupta et al., 2019) which features 1200+ different object categories. We report performances using the Average Precision (AP) metrics for rare classes, as previously established in the literature (Gu et al., 2022; Minderer et al., 2022). The open-vocabulary detector is initialized with the pretrained ViT backbone during finetuning. It adopts the simple feature pyramid and windowed attention to handle higher resolution images (e.g., 1024) as proposed in ViTDet (Li et al., 2022b), and Mask R-CNN heads and class-agnostic box/mask heads as in (Du et al., 2022; Gu et al., 2022; Zareian et al., 2021; Zhong et al., 2022). The model is trained with base categories, and tested to detect novel category objects at inference.

Table 5 presents our results on Open-Vocabulary detection which evaluates how well it does on detecting novel, previously unseen object categories. MaMMUT achieves 31.0 $AP_r$, which outperforms the existing ViT-based method OWL-ViT by 5.4 points. We also report AP performance for all classes for context, but note that our model is not optimized for detection AP, as is for example OWL-ViT which is a detection-only model.

## 6 Ablation studies

We here present ablation studies to understand the model characteristics and its design choices. We use a ViT-Base image encoder and a 128M language decoder, training on 4K batch size for 500K iterations unless noted otherwise.

**Cross-task benefits.** We first explore cross-task benefits between contrastive and text-generative pretraining. We find that joint training is generally favorable to tasks, but it affects tasks differently (Table 6). Compared to the contrastive-

| Method | APr | AP |
|---|---|---|
| DetPro-Cascade (Du et al., 2022) | 20.0 | 27.0 |
| Detic-CN2 (Zhou et al., 2022) | 24.6 | 32.4 |
| RegionCLIP (Zhong et al., 2022) | 22.0 | 32.3 |
| ViLD-Ensemble (Gu et al., 2022) | 21.7 | 29.6 |
| ViLD-Ensemble (Gu et al., 2022) | 26.3 | 29.3 |
| VL-PLM (Zhao et al., 2022) | 17.2 | 27.0 |
| Rasheed et al. (Rasheed et al., 2022) | 21.1 | 25.9 |
| OWL-ViT (Minderer et al., 2022) | 23.3 | **35.3** |
| OWL-ViT (Minderer et al., 2022) | 25.6 | 34.7 |
| **MaMMUT (ours)** | **31.0** | 32.8 |

Table 5: **Open-Vocabulary Detection Results.** MaMMUT performs well on detecting novel objects, scoring much higher on the Average Precision (AP) of rare objects $AP_r$.

| Contrastive | Generative | MS COCO | | Flickr30K | | VQA |
|---|---|---|---|---|---|---|
| | | I2T | T2I | I2T | T2I | Acc. |
| ✓ | | **54.8** | 38.2 | **82.6** | 67.1 | 63.5 |
| | ✓ | - | - | - | - | 69.9 |
| ✓ | ✓ | 54.3 | **38.7** | 80.6 | **67.5** | 71.7 |

Table 6: **Cross-task benefits.** Combining the contrastive and generative objectives yield benefits for generative task in our setup, while maintaining the performance of discriminative tasks.

only pretraining, joint training achieves signficantly better VQA performance by +8 points, which is expected. Joint training improves text-to-image retrieval. This is likely because the generative modeling enhances text representation, and while remaining very competitive on the image-to-text retrieval, it is not as beneficial, which can indicate potential competition of tasks (please see our later experiments which explore this). Compared to the generative-only pretraining, joint training outperforms on VQA by +1.8 points, likely because the contrastive learning helps to improve the joint image-text representation. We attribute this to better representation learning rendered from contrastrive training, also evidenced by other works (Radford et al., 2021; Jia et al., 2021). More importantly, joint training model is able to tackle retrieval tasks, not supported by the generative pretraining, and achieving satisfying performance overall.

**Cross-attention Design.** Another key exploration is the role of cross-attention mechanisms in the proposed joint training and how they affect various tasks. Cross-attention provide an efficient means for communication between the two modalities. We find that tasks indeed perform better under different circumstances. Specifically, cross-attention is preferred for text generative tasks, but not as much for contrastive ones. Table 7 shows that while denser cross-attention connections (e.g. 4) benefit the VQA task, one or few layers (e.g. 2) are sufficient for retrieval task.

**Balancing contrastive-vs-generative losses.** We here explore how to train the joint objectives, so that the tasks benefit fully from joint training. We do observe a potential competition of these tasks, thus a balance between losses needs to be obtained. Table 8 shows the performance trade-off when training jointly. The experiment is done on a smaller model and fewer steps, as mentioned above. We observe that the two objectives indeed have competitive behaviors. In the rest of the experiments, we pick equal weights over these two losses, as we observe that it gives more advantage to the VQA performance, whereas the retrieval does not suffer as much. This experiment provides an insight as to how to tune these parameters, depending on the requirements of the application.

**Scaling Image Encoder.** Effects of model scale on vision-language tasks have been explored in prior works (Alayrac et al., 2022; Chen et al., 2022). We present the results of scaling image encoder in Table 9, where we confirm increasing

| # Cross-Att. | MS COCO | | Flickr30K | | VQA |
|---|---|---|---|---|---|
| | I2T | T2I | I2T | T2I | Acc. |
| 1 | 55.3 | 39.6 | 81.7 | 67.2 | 68.7 |
| 2 | **56.6** | **40.1** | 81.9 | **67.6** | 70.8 |
| 4 | 55.7 | 39.9 | **82.2** | 67.3 | **71.5** |

Table 7: **Cross-attention Design.** Denser cross-attention layers are beneficial for the VQA task, whereas few cross-attention layers are sufficient for retrieval tasks.

| weights | MS COCO | | Flickr30K | | VQA |
| --- | --- | --- | --- | --- | --- |
| | I2T | T2I | I2T | T2I | Acc. |
| (2.0, 0.5) | **56.71** | **40.77** | **82.52** | **67.77** | 70.08 |
| (2.0, 1.0) | 56.7 | 40.27 | 81.84 | 67.25 | 70.84 |
| (1.0, 1.0) | 56.25 | 39.73 | 81.74 | 67.50 | 71.48 |
| (1.0, 2.0) | 55.39 | 39.09 | 81.54 | 65.64 | **72.27** |
| (0.5, 2.0) | 52.05 | 37.32 | 78.32 | 62.73 | 71.79 |

Table 8: **Balancing the losses.** Zero-shot retrieval results on COCO and Flickr30K benchmarks (R@1 shown) and VQA accuracy. Weight coefficients for the contrastive and generative loss are denoted as (constrastive, generative). We observe a clear trade-off between retrieval vs VQA tasks. We chose (1.0, 1.0) to balance the two objectives.

| Image Tower Size | MS COCO | | Flickr30K | |
| --- | --- | --- | --- | --- |
| | I2T | T2I | I2T | T2I |
| 86M | 62.0 | 44.2 | 84.6 | 71.1 |
| 300M | 66.4 | 49.4 | 91.2 | 78.8 |
| 630M | **70.7** | **54.1** | **94.9** | **82.5** |

Table 9: **Model scaling.** We show clear improvements on image-text retrieval with increasing image tower capacity.

the capacity of image encoder yields consistent improvement. We note that text encoder size of the last row has 1B parameters as opposed to 128M in the rest of the table. We use 16K batch size in this ablation.

**Video Model Ablations.** In Table 10, we compare the different design choices for adaptation of the video model. We explore the effects of removing gated connections, the fixed embeddings and even the video feature inputs. As seen, their importance increases in that order, where the addition of video tubes to process the video are of highest importance. This is not surprising as they learn the spatio-temporal video information. These ablations are done with the smaller ViT-L model.

# 7 Broader impacts

Developing vision-language models is very beneficial as they have many potential uses and applications, being able to understand both visual and language inputs better and to more accurately respond to questions. Some aspects of the model, specifically the text generative components might exemplify certain risks of generating off-topic, stereotypical, unwanted or other types of outputs, for which further investigation is needed. Our model has been trained on the same image-text dataset as many other previous works (Jia et al., 2021; Xiaohua Zhai & Beyer, 2022; Zhai et al., 2022; Yu et al., 2022; Yan et al., 2022) and is collected as noisy image-language pairs in a similar fashion to an even broader set of methods in the literature (Chen et al., 2022; Yuan et al., 2021; Alayrac et al., 2022). We developed the model for exploring novel research capabilities and have only used it for evaluation and visualization purposes in this paper.

# 8 Conclusions

We present the MaMMUT model which is a vision-encoder text-decoder model capable of multiple vision-language tasks. We propose two-pass learning which allows us to train jointly for retrieval and text-generative tasks with fully shared weights. The model is easy to adapt to video-language and object detection tasks. Our model accommodates a set of diverse tasks, such as image-text and text-image retrieval, novel (or open-vocabulary) object detection, VQA, VideoQA and Video Captioning, obtaining very strong performance on them with respect to the state-of-the art.

| | MSRVTT-QA | MSVD-QA |
| --- | --- | --- |
| MaMMUT- Full Model | **42.1** | **45.8** |
| No Gated Connection | 41.8 | 45.5 |
| No Fixed Embeddings | 41.5 | 45.1 |
| No Tubes | 40.3 | 42.6 |

Table 10: **Video Tubes Adaption Experiments.** Comparing different effects of adapting the model to video.

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

# A  Appendix

## A.1  Additional ablations

We here include additional ablation experiments which can provide further introspection into the model.

**Language pre-training effects.** As shown in this paper, a purely language model can be easily adapted for multiple image-language tasks. We here explore the effects of language pre-training on the vision-language tasks. Our results show that language-only pre-training is beneficial for image-text retrieval benchmarks, but the effects are minor (Table 11).

| | MS COCO | | Flickr30K | |
|---|---|---|---|---|
| pre-training? | I2T | T2I | I2T | T2I |
| No | 54.2 | **38.9** | 80.5 | 66.5 |
| Yes | **54.9** | 38.9 | **82.4** | **67.3** |

Table 11: **Effects of language-side pre-training.** Zero-shot image-text retrieval results on COCO and Flickr30K benchmarks, R@1 shown. Language pre-training provides small but consistent improvements.

**Projections and attention pooling.** Table 12 experiments with additional projections and attention pooling. We find that they are not needed for joint learning, indicating that the text decoder can be fully shared between tasks, which is a surprising finding. We only train for 100K iterations here.

**Bi-directional masking is important for contrastive learning.** Since here we are using a single language decoder to perform tasks of various characteristics e.g. retrieval and VQA, it is important to understand how to construct the language decoder for such tasks. We evaluated the effect of bi-directional or causal masking on a contrastive learning task on the Flickr dataset. We note that with these tasks, our model has to accommodate retrieval tasks which tend to process data with short text, whereas generative tasks tend to generate longer-text queries. We find that bi-directional masking benefits the contrastive learning much more than causal (Table 13), which is not surprising as the text associated with retrieval tasks is fully available before final feature representation is generated.

| Att. Pool/ Proj | image-to-text | | | text-to-image | | |
|---|---|---|---|---|---|---|
| | R@1 | R@5 | R@10 | R@1 | R@5 | R@10 |
| N/N | 53.32 | 80.57 | 86.91 | 39.86 | 67.7 | 77.46 |
| Y/N | 52.44 | 78.71 | 86.33 | 39.3 | 66.62 | 76.91 |
| N/Y | 51.37 | 78.81 | 86.13 | 39.49 | 66.97 | 77.05 |
| Y/Y | 50.78 | 76.95 | 85.74 | 38.59 | 66.54 | 76.07 |

Table 12: **Projection and attention pooling.** Our conclusions are that they are not needed to realize joint training. Retrieval performance on Flickr30K is shown.

| Masking | MS COCO | | Flickr30K | |
|---|---|---|---|---|
| | I2T | T2I | I2T | T2I |
| Causal | 53.0 | 37.5 | 78.1 | 62.5 |
| Bi-directional | **54.9** | **38.8** | **82.4** | **67.3** |

Table 13: **Bi-directional masking.** Contrastive learning clearly benefits from Bi-directional masking in the decoder language model.

## A.2 Total Train Compute Usage

We present the total compute used to train MaMMUT models in comparison with other foundational models by using the approximation technique in Brown et al. (2020). Some compute estimates are taken from Chen et al. (2022) e.g. PaLI (Chen et al., 2022), Flamingo (Alayrac et al., 2022), GIT-2 (Wang et al., 2022b) while others are estimated based on the paper details e.g. CoCa (Yu et al., 2022). The total train compute usage of MaMMUT is significantly lower than existing foundational models, for example, 3.4x cheaper than PaLI, 5.5x than CoCa, 10.3x than Flamingo, or 41.2x than GIT-2. Notably, MaMMUT is trained from scratch while some existing methods e.g. PaLI and Flamingo rely on pretrained image encoders either from contrastive learning or image classification.

