# OpenReview forum: "MaMMUT: A Simple Architecture for Joint Learning for MultiModal Tasks"
_TMLR — Accepted by TMLR_

### Review · Reviewer_dKme · 2023-04-14

**Summary Of Contributions:**

This work proposes MaMMUT as a simple paradigm of training with a decoder-only model for multimodal tasks. It can accommodate contrastive and generative learning by a novel two-pass approach on the text decoder. The effectiveness of the model is validated in a broad range of vision and language tasks such as image-text retrieval, video captioning, VQA, open-vocabulary detection tasks.

**Audience:**

No

**Broader Impact Concerns:**

No specific ethical concern in this work.

**Claims And Evidence:**

No

**Requested Changes:**

The task performance should be compared with recent SOTA models.


**Strengths And Weaknesses:**

<Strengths>

MaMMUT is simple and powerful enough to be applicable to various vision and language tasks with little modification.

<Weaknesses>

I think the main issue of this work is that it is outdated by recent progress of unified vision and language models.

- For example, this paper argues that “our model achieves the state of the art on image-text and text-image retrieval, video question answering and open-vocabulary detection tasks…”.

- However, it is not true any more; for example, BLIP-2 reports much better results.

- J. Li et al. BLIP-2: Bootstrapping Language-Image Pre-training with Frozen Image Encoders and Large Language Models, arXiv January 2023.

Given that the main strength of this work does not lie in the technical novelty itself, it should clearly show its empirical effectiveness. However, this work fails to validate it as ignoring comparison with many recent models.

---

### Review · Reviewer_Em6R · 2023-04-19

**Summary Of Contributions:**

The paper presents a unified architecture for performing image-text multimodal learning with both generative and contrastive objectives. The paper seeks to obtain the benefits of both types  of objectives under a single architecture by introducing a 2 pass approach to learning with both objectives. The authors perform extensive ablations and comparisons that show that their models produce reasonable gains over current state of the art approaches.

**Audience:**

Yes

**Broader Impact Concerns:**

No ethical concerns noted outside of the broader impacts presented in the paper.

**Claims And Evidence:**

Yes

**Requested Changes:**

1. I think the paper would benefit from having a section that discusses the key differences and similarities between their work and **CoCa**. As it stands, it seems that this work may be considered marginal relative to CoCa. Especially if the main claim of  " difficulty of extending to video tasks" is as a result of the use of an image-based backbone for CoCa instead of video based backbone like MaMMut uses.
2. Equation [1] is introduce but not used. It seems (correct me if I am wrong) that the it is the focal_constrastive loss that is eventually used. The section would be less confusing if Equation [1] were removed.


**Strengths And Weaknesses:**

## Strengths ##
1. Paper presents a unified architecture for doing both generative and contrastive  multimodal learning
2. The authors have extensive ablations that show the strength of their method


## Weaknesses ##
1. The proposed method of doing two passes seems quite computationally intensive and also has the potential to double the training runtime compared to previous approaches. This shortcoming is hardly discussed -- the authors do mention a 16% overhead compared to the pure contrastive approach -- but there is no elaboration of where this number comes from or what type of overhead is being referred to -- whether (a) compute in terms of flops (b) gpu memory (c) runtime
2.  An unexplored (possibly simpler) baseline that combines the two objectives without having to do two passes. Note that whilst the text embedding $l_i$  for the contrastive objective is obtained from pooling bi-directional representations, it should also be possible to pool representations that have been computed causally, but across the whole text -- this new embedding $\tilde{l_i}$ also contains global information across the whole text but now avoids the need to do two passes and also avoids having to force the model to do both bidirectional and causal learning. Cross attention with the image features can be done  beginning at later layers of the decoder (whilst the embeddings from the earlier half is pooled as suggested above and used for the contrastive objective.) So unlike CoCa, there is no need for 2 decoders.
3. One of the stated contributions is that the same architecture *Furthermore, the same architecture enables straightforward extensions to open-vocabulary object detection and video-language tasks* whilst previous approaches do not enable this. But from my reading of the paper, it seems this is just because previous approaches do not use the video based back-bone that this paper does. **Can you confirm that there is something else fundamental to your method, outside of these other approaches, that allows you to use a video based back bone ?**



### A Few Nits ###
1.  From the abstract -- *The development of language models have moved from encoder-decoder to decoder-only designs*.  Language models have always primarily been decoder only models so this statement is not accurate. It is repeated several places in the paper.
2. The statement that the two types of objectives conflict is repeated multiple times in the paper without citation or an ablation to justify.

---

### Review · Reviewer_GqnS · 2023-05-01

**Summary Of Contributions:**

This paper presents a novel multimodal modeling approach that combines contrastive learning, localization-aware pretraining, and autoregressive captioning in a single model, utilizing an image encoder and text decoder with a two-pass learning strategy. The results demonstrate state-of-the-art performance on various benchmarks for models of similar size. However, there is potential for more robust ablations to yield more conclusive results.



**Audience:**

Yes

**Claims And Evidence:**

Yes

**Requested Changes:**

1. Most vision and language works only report model parameters while training on datasets with differing sizes or characteristics. A shift towards reporting total compute used(as is used in most LLM papers nowadays), which takes into account dataset size and training duration, would be beneficial.
2. Open Vocabulary Detection results: Given the enhanced performance in rare categories, what is the influence of pretraining dataset on these results? Does Owl-ViT use a similar/same pretraining dataset?
3. For ablations regarding balancing contrastive-vs-generative losses, the authors should explore more data points by adjusting the weights. Comparisons between two weighting schemes do not seem sufficient for drawing conclusions regarding the hypothesis.
4, Table 9's ablations on model size effects compare various models with differing sizes. For a more rigorous ablation, only one hyperparameter should be altered while maintaining all others constant. The validity of conclusions drawn when comparing different models trained on different datasets is uncertain.

**Strengths And Weaknesses:**

## Strengths

1. The incorporation of both contrastive and generative learning within a single decoder offers a streamlined approach.
2. The model's adaptability to detection and video classification tasks lends it a degree of generality.
3. The robust performance across multiple tasks in comparison to state-of-the-art methods supports the validity of the proposed approach and pretraining strategy.

## Weaknesses

1. It is unclear how training from scratch compares with initializing from a pretrained VIT. Or similarly from a pretrained language decoder. Is training from scratch necessary?
2. VQA2 results : Although the inclusion of generative pretraining alongside contrastive losses appears advantageous, comparisons with Flamingo and PaLI are rendered uncertain due to differing datasets. Consequently, it is difficult to determine whether improvements stem from the specific pretraining approach and architectural changes or due to larger/better quality datasets.
3. Table 7's ablations do not seem to provide definitive evidence that fewer cross-attention layers are unsuitable for retrieval. While MS COCO retrieval figures decline, Flickr retrieval numbers improve.
4. Ablations in Table 9 that compare model sizes across models trained on different datasets is not a conclusive approach in my opinion.

---

### Decision · Action_Editors · 2023-07-08

**Recommendation:** Accept as is

**Comment:**

This paper presents a novel multimodal modeling approach. All three reviewers are positive about the work due to the novel integration of generative and contrastive learning as well as strong results across multiple tasks and datasets.

During the initial reviews, the reviewers have concerns regarding
- technical details (compute use estimation, clarification about Ablations)
- additional ablation (e.g., balancing contrastive-vs-generative losses)
- comparisons with relevant work (e.g., CoCa and BLIP-2)

The authors' responses and paper revision sufficiently addressed these concerns. All reviewers vote for "Learning Accept". The AE reads the reviews, authors' responses, and the paper and agrees with the reviewers that the paper meets the bar for TMLR. The AE thus recommends to accept.




**Audience:**

Yes, multimodal learning is of great interest to the community.

**Claims And Evidence:**

Yes, the claims are accurate and well supported by the extensive ablation study.